# Synthetic Peptide ΔM4-Induced Cell Death Associated with Cytoplasmic Membrane Disruption, Mitochondrial Dysfunction and Cell Cycle Arrest in Human Melanoma Cells

**DOI:** 10.3390/molecules25235684

**Published:** 2020-12-02

**Authors:** Gloria A. Santa-González, Edwin Patiño-González, Marcela Manrique-Moreno

**Affiliations:** Structural Biochemistry of Macromolecules Group, Chemistry Institute, Faculty of Exact and Natural Sciences, University of Antioquia, Medellin A.A. 1226, Colombia; gloriasanta@itm.edu.co (G.A.S.-G.); edwin.patino@udea.edu.co (E.P.-G.)

**Keywords:** melanoma skin cancer, antimicrobial peptides, antiproliferative peptides, cell cycle arrest, membrane integrity

## Abstract

Melanoma is the most dangerous and lethal form of skin cancer, due to its ability to spread to different organs if it is not treated at an early stage. Conventional chemotherapeutics are failing as a result of drug resistance and weak tumor selectivity. Therefore, efforts to evaluate novel molecules for the treatment of skin cancer are necessary. Antimicrobial peptides have become attractive anticancer agents because they execute their biological activity with features such as a high potency of action, a wide range of targets, and high target specificity and selectivity. In the present study, the antiproliferative activity of the synthetic peptide ΔM4 on A375 human melanoma cells and spontaneously immortalized HaCaT human keratinocytes was investigated. The cytotoxic effect of ΔM4 treatment was evaluated through propidium iodide uptake by flow cytometry. The results indicated selective toxicity in A375 cells and, in order to further investigate the mode of action, assays were carried out to evaluate morphological changes, mitochondrial function, and cell cycle progression. The findings indicated that ΔM4 exerts its antitumoral effects by multitarget action, causing cell membrane disruption, a change in the mitochondrial transmembrane potential, an increase of reactive oxygen species, and cell cycle accumulation in S-phase. Further exploration of the peptide may be helpful in the design of novel anticancer peptides.

## 1. Introduction

Cancer is a multifactorial and heterogeneous disease characterized by genetic modifications in normal cells, as a result of which they become malignant. These transformed cells are characterized by uncontrolled cell cycle progression, the evasion of programmed cell death, and an invasive capacity [1,2]. Skin cancer is one of the forms of cancer with the highest incidence in the Caucasian population, with 2–3 million new cases reported per year worldwide. Moreover, incidence rates are expected to be higher in the coming decades [3]. Melanoma, which is the most serious type of skin cancer, is a neoplasm of the cells that produce melanin (melanocytes)—the pigment that gives the skin its color. Melanoma can also manifest in the eyes and, rarely, in internal organs, such as the intestines. It is known as the deadliest form of skin cancer, because it has a tendency to spread. General risk factors for this pathology include prolonged exposure of the skin to ultraviolet (UV) radiation, including sunlight and tanning beds; skin pigmentation; epidermic lesions such as moles; genetic factors; and a compromised immune system [4]. Current treatments include surgery, which can result in significant disfigurement, and radiotherapy, which has severe side effects, leading to emotional and physical consequences for patients. It has been reported that tumor cells can develop a form of resistance to conventional chemotherapeutics whereby the drugs are pumped out by multi-drug-resistant proteins [5], which also facilitate repair of DNA damage, the tolerance of stress conditions, and an abnormal expression of drug detoxifying enzymes [6]. Therefore, efforts to design and evaluate novel molecules for the potential treatment of skin cancer are necessary.

Antimicrobial peptides (AMPs) are considered part of the innate immune system of several organisms and participate in the first line of defense in response to the attack of pathogens, such as Gram-negative and -positive bacteria, envelope viruses, fungi, and parasites [7,8,9]. Most AMPs are comprised of 50 residues or less, are amphipathic, and have a net positive charge at a physiological pH [10]. They are small, relatively easy to synthesize and modify, and capable of penetrating cell membranes. They also have a high potency of action and a wide range of targets which may help to reduce their harmful side effects [11,12,13,14]. A promising application for emerging AMPs is as therapeutic agents for various pathologies, as peptides have important advantages compared to other medicinal molecules [15,16]. Approximately 3000 natural AMPs have been isolated and characterized from bacteria, protozoa, protists, fungi, plants, and animals according to the Antimicrobial Peptide Database (APD, http://aps.unmc.edu/AP/main.php) [17]. From these 3000 AMPs, 230 also possess anticancer activity and are classified as anticancer peptides (ACPs) [17]. For this reason, the evaluation of novel ACPs, either alone or in combination with other conventional drugs, has been regarded as a potential strategy to be explored [18]. Current ACPs have been divided into three major groups: (a) Antimicrobial/pore-forming peptides; (b) cell-permeable peptides; and (c) peptide-targeting tumors [19]. The mechanism of action of ACPs that target cell membranes is based on electrostatic interactions between the cationic residues on the peptide and anionic lipids on cancer cell membranes [12,20]. However, ACPs also target different cellular structures, such as mitochondria [21,22,23], and interfere with the transduction pathway and cell cycle [11].

Several studies have evaluated the effect of ACPs on skin cancer cells, with promising results [24,25,26,27,28,29]. However, the therapeutic targets have been independently studied, and in some cases, it has been necessary to modify the original sequences in order to increase the anticancer activity and decrease the side effects. Evaluations of new ACPs and studies on their interactions with mammalian cells are important for advances to be made in this field that contribute to the design and development of new drugs for the treatment of cancer. In previous studies, we focused on designing and evaluating synthetic peptides, such as ΔM2. This is a Cecropin D-like analog with a +9 charge at pH 7.4 that is active against Gram-negative and Gram-positive bacteria [30], but not against cancer cells. After further modifications of the N-terminal fragment of ΔM2, it became ΔM4—a 20-residue AMP—which, at a physiological pH, has a charge of +7. In the present study, the antiproliferative activity of the synthetic peptide ΔM4, as well as its possible multitarget action on human melanoma cells (A375) and spontaneously immortalized human keratinocytes (HaCaT), was investigated. The effects of ΔM4 treatment regarding the cell membrane integrity, mitochondrial function, and cell cycle progression were evaluated.

## 2. Results

### 2.1. Design and Prediction of the 3D Structure of ΔM4

∆M4 is a synthetic 20 residue AMP with a +7 charge in physiological conditions. This fragment contains almost all the amino acids related to the global charge of the parent peptide ∆M2, which is a 39 residue peptide with a +9 charge at pH 7.4, and a hydrophobicity of 46.3%. ∆M2 was previously designed and synthesized by us, and showed a high selectivity towards anionic bacterial membranes [30], but not against cancer cells (data not shown). The design of ∆M4 included the substitution of four amino acids in the N-terminal fragment of ∆M2: G11, R15, and A17 to W, and I19 to Y. The substitutions were selected with the aim of increasing the hydrophobicity of ∆M4 (52.1%) in comparison to the parent peptide. According to the I-TASSER results, the prediction of the secondary structure of ∆M4 shows a single α-helix (Figure 1a). Figure 1b presents the helical wheel projection of the ∆M4 peptide. It shows an amphipathic structure with two faces: One conserving the hydrophobic residues and the other forming a polar surface. It is known that AMPs are usually in a random state in aqueous environments, but generally fold during their interaction with membranes.

### 2.2. ΔM4 Disrupts the Plasma Membrane and Selectively Reduces Melanoma Skin Cancer Cell Viability in a Dose-Response Relationship

A common marker for cell death is the membrane integrity. To determinate the cytotoxic effect of ΔM4, the A375 melanoma cell line and the control non-tumoral cell line HaCaT were treated with the peptide and analyzed by flow cytometry. The results in Figure 2a show that A375 and HaCaT cells responded differently to treatment with ΔM4. After 24 h of treatment, ΔM4 induced a cytotoxic effect in human melanoma cells, with a significant decrease in cell viability at concentrations of 25 μM and higher (*p* ≤ 0.0001). In the non-tumoral cells, 24-h exposure to ΔM4 had a limited effect; the viability was reduced at doses of 25 μM (*p* ≤ 0.05), with an effect that remained invariable at higher concentrations. Figure 2b shows representative histograms of propidium iodide (PI) uptake in A375 melanoma cancer cells analyzed by flow cytometry. In the histograms, it is evident that the treatment with ΔM4 for 24 h induced a distinctive and dose-dependent increase in PI fluorescence, indicative of cytoplasmic membrane permeabilization.

Table 1 shows the half-maximal inhibitory concentration (IC_50_) determination of the peptide ∆M4 in both cell lines. The results revealed an IC_50_ value of 9.31 μM for A375 melanoma cancer cells and 88.56 μM for non-malignant HaCaT cells. A selectivity index (SX) [31] was calculated as $\mathrm{SX}=\frac{{\mathrm{IC}}_{50}\;\mathrm{HaCaT}}{{\mathrm{IC}}_{50}\;A375}\times 100$, and the value obtained was 951.3, which is indicative of selective toxicity to melanoma skin cancer cells.

### 2.3. ΔM4-Induced Morphological Changes in Melanoma Skin Cancer Cells

To investigate the mode of action of ΔM4 on melanoma cells, morphological changes were examined by differential interference contrast (DIC) microscopy and quantified by flow cytometry according to forward scatter/side scatter (FSC/SSC). The results indicated that A375 cells treated with ΔM4 had a lower relative size and increased intracellular granularity compared to the untreated control. In Figure 3a, representative images obtained by differential interference contrast (DIC) microscopy are shown. A typical elongated epithelial morphology, with a wide cell spread on the growth surface and round cells slightly adhering to the substrate (mitotic) were observed in the untreated (control) melanoma cells. Furthermore, cells did not present intracytoplasmatic granules, and no cell lysis or reduction of cell growth was evident. Treatments with 25 μM of ΔM4 induced changes in the cellular morphology; lysed cells were present, and the formation of vacuoles and growth inhibition were observable. ΔM4 at 50 μM provoked deleterious effects; more than 70% of the cells were round and loosely attached, intracytoplasmatic granules were more evident, extensive cell lysis was observable, and cellular debris was present. Treatment with ΔM4 at 75 μM induced severe morphological changes in A375 cells. In the image, it is possible to observe rounded or lysed cells, the destruction of cell monolayers, complete growth inhibition, and cellular debris extended on the growth surface.

Morphological features were quantified by flow cytometry analysis. In Figure 3b, it can be observed that the size of A375 cells is significantly reduced with ΔM4 treatment compared to the untreated cells (*p* = 0.0034). Additionally, Figure 3c shows a dose-dependent relationship between the ΔM4 peptide concentration and cellular complexity in cancer melanoma cells, with a significant increase in intracellular granularity at 75 μM (*p* = 0.0239, with respect to untreated cells).

### 2.4. Mitochondrial Membrane Hyperpolarization and ROS Production Are Stimulated in A375 Cells by ΔM4 Treatment

A reported mechanism of the anticancer activity of AMPs is the induction of cell apoptosis through mitochondrial interactions [32,33]. To further investigate the involvement of mitochondria in the mechanism by which ∆M4 induces cell death, the membrane-permeable lipophilic cationic fluorochrome 3,3′-dihexyloxacarbocyanine iodide (DiOC_6_(3)) was used to monitor the mitochondrial membrane potential changes after the peptide treatment. As shown in Figure 4a, the exposure of melanoma cancer cells to different ∆M4 concentrations significantly increased the mitochondrial DiOC_6_ intake at 50 and 75 μM (*p* = 0.0454). These results suggest a high trans-membrane potential, known as hyperpolarization. This could be related to the reduction in cell viability observed in previous experiments.

Due to its high mitochondrial potential, the mitochondrial respiratory chain becomes a significant producer of reactive oxygen species, which can enhance the deleterious effect of the peptide treatment. Therefore, the mitochondrial changes in ROS levels after ΔM4 treatment were examined using MitoTracker Red CMXRos, which is a red-fluorescent dye that stains mitochondria in live cells and whose accumulation is dependent upon the membrane potential. Representative histograms obtained in flow cytometry and quantification of the mean fluorescence intensity are shown in Figure 4b. An increased mitochondrial ROS concentration was observed in A375 cells at all ∆M4 concentrations evaluated, in comparison with untreated cells. Statistical significance was obtained for 50 μM (*p* = 0.0378) and 75 μM (*p* = 0.0130b).

### 2.5. Exposure to ∆M4 Induces Cell Cycle Arrest at the S-Phase in A375 Cells

A further investigation to characterize the antiproliferative effect of ΔM4 in A375 cells was carried out by analyzing the cell cycle distribution (Figure 5a). Analysis by flow cytometry of A375 cells in the absence of ΔM4 revealed a typical cell cycle distribution, with most cells in the G1 phase (62.66%). Compared to untreated cells, ΔM4-treated A375 cells exhibited a significant decrease in the percentage of cells in the G1 phase at 50 and 75 μM of the peptide (38.63% and 64.30%, respectively). This decrease in cells in the G1 phase is concomitant with an S-phase accumulation (from 26.17% in untreated cells to 34.28% at 75 μM of ΔM4), followed by the appearance of a sub-G1 peak.

As few peptides that target the cell cycle have been reported, cell cycle distribution assays were extended to the non-tumoral cell line HaCaT, in order to evaluate different cell responses (Figure 5b). The results showed that 24 h exposure to ΔM4 had little or no significant impact on the cell cycle distribution in non-tumoral cells.

## 3. Discussion

Skin cancer is one of the tumors with the greatest incidence in humans. This disease affects millions of people, and epidemiological studies have predicted an alarming increase in the number of new cases in the coming years [29]. Therefore, it is necessary to evaluate the potential use of antiproliferative agents selective to tumor cells. This study evaluated the antiproliferative effect of the synthetic peptide ΔM4, as a potential ACP. ∆M4 is a +7 charged peptide at a physiological pH. The residues in the sequence were substituted with tryptophan in order to increase the hydrophobicity of the peptide as a result of the well-known indole ring of the tryptophan to anchor in the polar-apolar interface of the lipid membranes [34,35]. Tyrosine has also been found to display significant interactions with phospholipid headgroups of membranes [36,37]. The helical wheel projection of ∆M4 shows an amphipathic structure, of which one faces contains the residues associated with the charge of the peptide, and the other contains the hydrophobic residues. It has been proposed that the first stage of all ACP mechanisms is based on electrostatic interactions between the positively charged amino acids of the peptides and the anionic groups of the target cellular membrane. This interaction is the basis on which these molecules are considered to be more selective towards tumor cells, given that there is a difference in composition between tumor and non-tumor cells [19].

However, there have been limited evaluations of new ACPs and the study of their interaction with mammalian cells. As such, advances in this area of research would help facilitate the design and development of new compounds with potential as drugs for skin cancer treatment. In this study, how the peptide exercises its antiproliferative function in cell lines was explored through various in vitro techniques. To achieve the objective of the study, the cytotoxicity of the peptide in the human cell line of A375 skin cancer and non-tumoral HaCaT cells was evaluated. The tumor cells showed a significant decrease in viability at all of the peptide concentrations evaluated. Moreover, at the maximum concentration of ΔM4, a 95% reduction in the viability of the tumor cells was induced, while the viability of the non-tumor cells was not affected to the same extent. This differential effect of the peptide in the two cell lines was evidenced by the IC_50_, which was approximately 10 times lower for the A375 cells (IC_50_ = 9.31) than that obtained for the HaCaT cells (IC_50_ = 88.56). Evidence exists that, as proposed by Alves et al., the plasma membrane of tumor cells exhibits important differences to that of non-tumor cells, including the phospholipid composition, the pH of the extracellular medium, the surface charge, and the fluidity of the membrane [17]. Concerning the composition, the external monolayer of the non-tumor cells is principally composed of phosphatidylcholine and sphingomyelin, which are zwitterionic phospholipids that do not provide a charge to the surface of the non-tumor cells. However, numerous studies have shown that, unlike non-tumor cells, cancer cells contain high levels of phosphatidylserine, heparin sulfate, sialylated gangliosides, and *O*-glycosylated mucin in the external face of the cellular membrane [30,31,32,33,34]. This change in the composition favors the interaction of cationic peptides with the surface of the external monolayer of the membranes of the cancer cells, making them more susceptible to the first phase of electrostatic attraction between ACPs and tumor membranes. This promotes the specificity of ACPs toward cancer cells, without them being affected by tumor heterogeneity [38], which broadens the potential use of the peptide to other tumoral cell lines. Ma et al. studied three kinds of tumor cells: The breast cancer cell line, MCF-7; the malignant melanoma cell line, A375; and the brain glioma cell line, U87. These were treated with the ACPs, AP1-Z1, and its six mutants. A similar trend in the antiproliferative effects on the different tumor cells was achieved, as shown by the 3-(4,5-dimethylthiazol-2-yl)-2,5-diphenyltetrazolium bromide or MTT results. [39].

The peptide ΔM4 has a cytotoxic effect that is dose-dependent and acts with selectivity for tumor cells. After the cells are exposed to ΔM4, the A375 cells present an elevated membrane permeability, which is ultimately reflected in an increase in the cytotoxicity. These results highlight the potential of the peptide to inhibit the viability of cancer cells. The ΔM4 peptide has a significant effect on A375 tumor cells, reducing the cell size and increasing the granularity. These changes in the cell morphology are indicative of a compensatory response induced by stimuli of cell stress. When these response mechanisms are not sufficient for homeostasis to be recovered, permanent damage is caused to the cell structures, inducing cell death [35]. The morphological characteristics observed in melanoma cells exposed to the peptide demonstrate an increase in vacuolization of the cytoplasm that varies according to the concentration. In tumor cells, vacuolization is commonly associated with the response to chemotherapeutic agents and low-molecular-weight compounds. Vacuolization often accompanies regulated cell death (RCD); however, its role in cell death processes remains unclear [40,41,42].

Variation in the potential of the mitochondrial membrane is one of the main markers associated with programmed death [39,40]. Given that the objectives of this study include the identification of how ∆M4 exerts its antiproliferative action, the effect of the peptide on this cellular organelle was studied. In melanoma cancer cells, exposure to different ∆M4 concentrations increases the mitochondrial DiOC_6_ intake, suggesting a high trans-membrane potential, namely hyperpolarization of the inner mitochondrial membrane, which can eventually lead to the release of apoptotic factors to activate the intrinsic apoptosis pathway [43,44]. These results are in accordance with those reported in other studies, which show that peptides that induce cytotoxicity by different mechanisms not only directly attack the plasma membrane, but can also cause lesions in other cellular organelles, particularly mitochondria [19]. Dysfunctional mitochondria are a source of ROS, mainly through the electron transfer chain (ETC), which generates an escape of electrons that react with oxygen (O_2_) and form O_2_^−^ free radicals. The dismutation of superoxide anion (O_2_^−^) results in the formation of hydrogen peroxide (H_2_O_2_) and hydroxide anion (OH^−^), which are both capable of inducing damage to critical cell structures [45,46], possibly increasing the cytotoxic effect of the peptides. The results show that, when the A375 cells were exposed to the ∆M4 peptide, the concentration of mitochondrial ROS increased at all of the peptide concentrations evaluated. Interestingly, it has been reported that the use of other compounds with chemotherapeutic potential directed against mitochondria induces dysfunction of the organelle and catastrophic vacuolization in cancer cells [39]. As mitochondria play an important role in the physiology, viability, and proliferation of tumor cells, the dysfunction associated with treatment with ∆M4 could be explained in part by the cytotoxic effect of the peptide, which could affect mitochondrial homeostasis and important cell processes, such as the respiratory capacity, energy production, autophagy, apoptosis, and the cell cycle [45].

Given that many chemotherapeutics affect cancer cells by altering the cell cycle, generally in specific points of control, the next stage in this study was to test whether exposure to ΔM4 affected the cycle of A375 cells. Flow cytometry analysis revealed that the peptide caused an accumulation of tumor cells in phase S, together with the appearance of a small number of cells in the subG1 stage. This suggests that ΔM4 causes the arrest of tumor cells at the S/G2 transition, which could contribute to the inhibition of cellular growth. The accumulation in the S stage could be associated with DNA damage and the response to replication stress, which causes the cell to stop its normal cycle in order to attempt to repair the genetic material. However, further study is required to test this hypothesis. Cyclin proteins and cyclin-dependent kinases (CDK), as well as other DNA molecules, damage sensors and regulate the progression of the cell cycle, controlling the temporal order of each of these phases [46,47]. Some authors have reported that, when cells undergo DNA damage and accumulation as the phase S occurs, the levels of the different proteins, including CDK2 [47], CDK1, and ATR [48], are increased. However, the mechanisms of control that regulate the S/G2 transition are not clearly understood. Various highly-used chemotherapeutic agents, such as cisplatin, doxorubicin, and 5-fluorouracil (5-FU), have shown the capacity to interact with DNA or block synthesis, which causes the accumulation of tumor cells in phase S, in turn leading to cell death. Moreover, it has been found that there are no significant changes in the distribution of the cell cycle phases of HaCaT non-tumor cells after treatment with the peptide, further demonstrating the effective selectivity of ΔM4 on non-tumor cells.

Natural ACPs with high anticancer activity usually have a sequence of more than 30 amino acids, which greatly increases the cost of synthetic production [49]. Currently, only a small number of ACPs have entered clinical trials, such as ITK-1 (Green Peptide), ACG-1005 (AngioChem), and MBI-226 (Cadence Pharmaceuticals), since the synthesis cost of these is higher than that of organic small molecule drugs [49]. The results obtained in this study demonstrate that the synthetic 20 residue ΔM4 has significant and differential cytotoxicity on skin cancer cells. The IC_50_ values obtained in tumor cells are in the range considered appropriate for therapeutic use and present significant differences with respect to HaCaT control cells. While other studies have evaluated the use of different classes of peptides on skin cancer cells with promising results, therapeutic targets have been studied independently. Do et al. [24] studied the cytotoxic effect of the peptide cationic Melittin on non-melanoma skin cancer cells and described its action as a membrane-disrupting agent that ultimately induces cell death. Brown and collaborators studied the effect of the peptide disintegrin obtustatin on melanoma cells and reported its action as an inhibitor of α1β1 integrin, resulting in blocked angiogenesis and tumor growth in animal models [50]. Camilio et al. [28] evaluated the antitumoral effect of the peptide LTX-315 on murine melanoma cells and other animal models, with results demonstrating that treatment with the peptide induces tumoral necrosis initiated by a disruptive effect on the plasma membrane of the tumor cells. Eike et al. [22] reported, in a separate study, that LTX-315 not only acts on the membrane, but also affects the mitochondria and induces the liberation of danger-associated molecular pattern molecules (DAMP), which in turn induce a systemic immune response in tumor cells in animal models of melanoma. Recent work by Santa-Gonzalez et al. [51] studied the effect of LTX-315 on cell cycle progression in A375 melanoma cells, concluding that treatment with the peptide does not significantly affect the distribution of cells in the cell cycle phase. Meanwhile, Marquez et al. [52] described the cytotoxic and antiproliferative effect of crude skin secretion from *Physalaemus nattereri* on B16F10 murine melanoma cells, finding evidence of cell death caused by apoptosis and an apparent arrest in the S-phase of the cell cycle, which is an affect that they ascribed to anti-cancer molecules related to antimicrobial peptides present in the frog secretions.

This study reports a unique, unmodified, and highly selective sequence for cancer cells, which could affect the cytoplasmic membrane, mitochondria, and cell cycle, enhancing its antitumoral effect. The results showed that the effectiveness of ΔM4 is comparable to the cytotoxicity reported for the peptide LTX-315, although the mechanisms appear to be different. It has been reported that the main form of action of the peptide LTX-315 is the induction of cell lysis, while ΔM4 causes a functional alteration of the mitochondria and appears to promote apoptosis. However, further study is required to confirm this hypothesis, potentially enabling the use of the ΔM4 sequence as an antiproliferative peptide.

## 4. Materials and Methods

### 4.1. Prediction of the Peptide Structure

A prediction of the 3D structure of the ∆M4 peptide was generated using two types of software: I-TASSER V5.1 was used to generate the atomic model through the identification of structural templates from the PDB using LOMETS, and the structure was predicted by multiple threading alignments and iterative structural assembly simulation (https://zhanglab.ccmb.med.umich.edu/I-TASSER/). The atomic model was refined using ModRefiner to obtain a peptide structure closer to its native form (https://zhanglab.ccmb.med.umich.edu/ModRefiner/) [53]. The helical wheel projection was calculated using the Net Wheels application (http://ibqp.unb.br/NetWheels) [54].

### 4.2. Peptide Synthesis

ΔM4 (NFFKRIRRAWKRIWKWIYSA, Lot. b88771380001/PE2074) was synthesized by the solid-phase method, and purchased from GenScript (Piscataway Township, NJ, USA). The purity of the peptide was determined to be higher than 95% by analytical HPLC, TFA removal was performed, and the molecular weight was confirmed with MALDI-TOF mass spectrometry.

### 4.3. Cell Culture

Human melanoma A375 cells (ATCC, CRL-1619) and the non-tumoral human keratinocyte cell line HaCaT were cultured in Dulbecco’s modified Eagle’s medium (DMEM) completed with 5% fetal calf serum, 100 µg/mL penicillin, and 100 µg/mL streptomycin, and stored in a humidified incubator at 37 °C with 5% CO_2_/95% air. Cell cultures were periodically checked under a microscope to ensure a normal morphology, adhesion, and subcultures before reaching monolayer confluence.

### 4.4. Treatment Conditions

Melanoma and non-tumoral skin cells were seeded in 6-well plates at a concentration of 2.5 × 10^5^ cells/mL. Cells were cultured and stored under culture conditions described previously. After a time-lapse of 24 h to ensure adhesion and exponential growth, cells were treated with concentrations of 25, 50, and 75 μM of the ∆M4 peptide for 24 h and subsequently processed for different tests. All data presented in this report represent results obtained in at least three independent experiments per treatment group.

### 4.5. Evaluation of the Cytoplasmic Membrane Integrity as a Measure of the Cell Viability

For assays of the membrane integrity as an indicator of cell death, cells were stained with propidium iodide (PI) to check the incorporation of the dye. After 24 h of ∆M4 peptide treatment, cells were washed twice in phosphate buffered saline, trypsinized, pelleted, and dyed with 1 mg/mL PI (Sigma, St. Louis, MO, USA, P4170). Following this, 10,000 events were acquired by flow cytometry in BD LSRFortessa. The mean fluorescence intensity (MFI) was evaluated utilizing FlowJo V7.6. Live and dead cells were gated, and the absolute IC_50_ was calculated with GraphPad Prism V6.

### 4.6. Morphological Analysis

A375 cells were seeded in microplates and cultured under normal culture conditions. Once cell adhesion was achieved, the peptide ∆M4 was added at different concentrations. After 24 h of treatment, the cells were washed and prepared for observation and photographing under differential interference contrast (DIC) microscopy.

### 4.7. Cell Size and Granularity, and Cell Cycle Analysis by Flow Cytometry

A375 melanoma cells were exposed to the ∆M4 peptide for 24 h and subsequently collected and centrifuged. The cell pellet was fixed in 70% cold ethanol for 1 h. Permeabilized cells were incubated with 100 μg/mL of RNase (Sigma, St. Louis, MO, USA, R5000), labeled with 100 μg/mL of propidium iodide (Sigma, P4170) for 30 min, and analyzed by BD LSRFortessa flow cytometer (Franklin Lakes, NJ, USA). Using the forward scatter (FSC) and side scatter (SSC) parameters, the relative size and granularity of the cells were determined. PI fluorescence was used for cell cycle analysis, and the phase distribution was calculated in FlowJo V7.6.

### 4.8. Evaluation of the Mitochondrial Membrane Potential

A375 cells were seeded and prepared according to treatment conditions. After incubation with ∆M4, cells were assayed using DiOC_6_ (Molecular Probes, Eugene, OR, USA, D273). Cells were suspended in phosphate buffered saline containing 50 nM of DiOC_6_ and 1 mg/mL of PI (Sigma P4170). Cell suspensions were protected from light and stored at room temperature for 20 min. Afterward, cells were washed and analyzed by BD LSRFortessa flow cytometrer (Franklin Lakes, NJ, USA). For analysis, dead cells were excluded, and the mean fluorescence intensity (MFI) of DiOC_6_ was evaluated utilizing FlowJo v7.6.

### 4.9. Mitochondrial ROS Detection

Relative levels of mitochondrial ROS were measured using MitoTracker Red (Invitrogen, Carlsbad, CA, USA, M7512). After treatments, A375 cells were exposed to 3 μM of dye for 15 min, incubated at 37 °C, washed twice in phosphate buffered saline, and analyzed using a BD LSRFortessa flow cytometer. Results were expressed as the mean fluorescence intensity (MFI) of Mitotracker.

### 4.10. Statistical Analysis

Statistical tests and graphs were performed and developed using GraphPad Prism V6. All data represent results obtained from three independent experiments per treatment group. Comparisons of data were carried out with ANOVA analysis, followed by Fisher’s protected least significant difference (FPLSD) tests. Data were expressed as the mean ± standard error of the mean (SEM). *p* ≤ 0.05 was considered statistically significant.

## Figures and Tables

**Figure 1 molecules-25-05684-f001:**
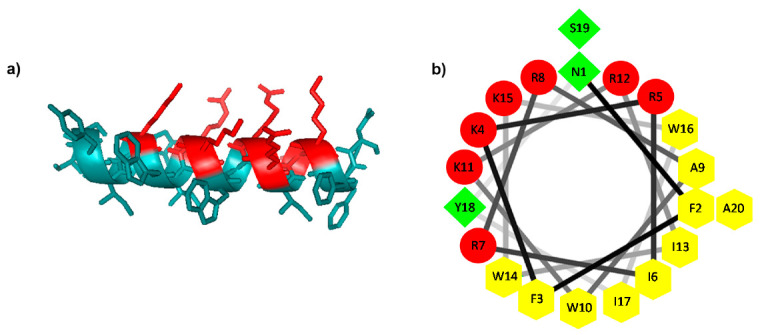
Predicted structure of the ∆M4 peptide. (**a**) Predicted α-helical structure of the ∆M4 peptide; positively charged residues are highlighted in red. (**b**) Helical-wheel projection of ∆M4; basic residues are presented as pentagons, non-polar residues as green squares, and polar uncharged residues as circles.

**Figure 2 molecules-25-05684-f002:**
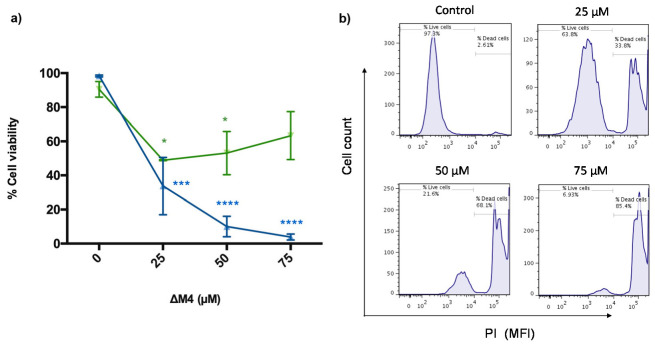
Selective cytotoxic effect of ΔM4 in A375 cells. (**a**) Tumoral (▲) and non-tumoral cells (▼) were treated with different concentrations of peptide for 24 h, before being dyed with propidium iodide (PI) and analyzed by flow cytometry. The percentage of viable cells in tumoral A375 and non-tumoral HaCaT cell lines was determined by PI staining, where live cells exclude the dye and dead cells are positive for it. Values are expressed as the mean ± standard error of the mean (SEM) of three independent experiments. Two-way ANOVA presented the difference with respect to non-treated cells, where * *p* ≤ 0.05, *** *p* ≤ 0.001, and **** *p* ≤ 0.0001. (**b**) Membrane-permeabilization activity of the ΔM4 peptide in tumoral A375 cells assayed for PI uptake by flow cytometry. The histograms show a representative example of the mean fluorescence intensity (MFI) of the dye in each treatment.

**Figure 3 molecules-25-05684-f003:**
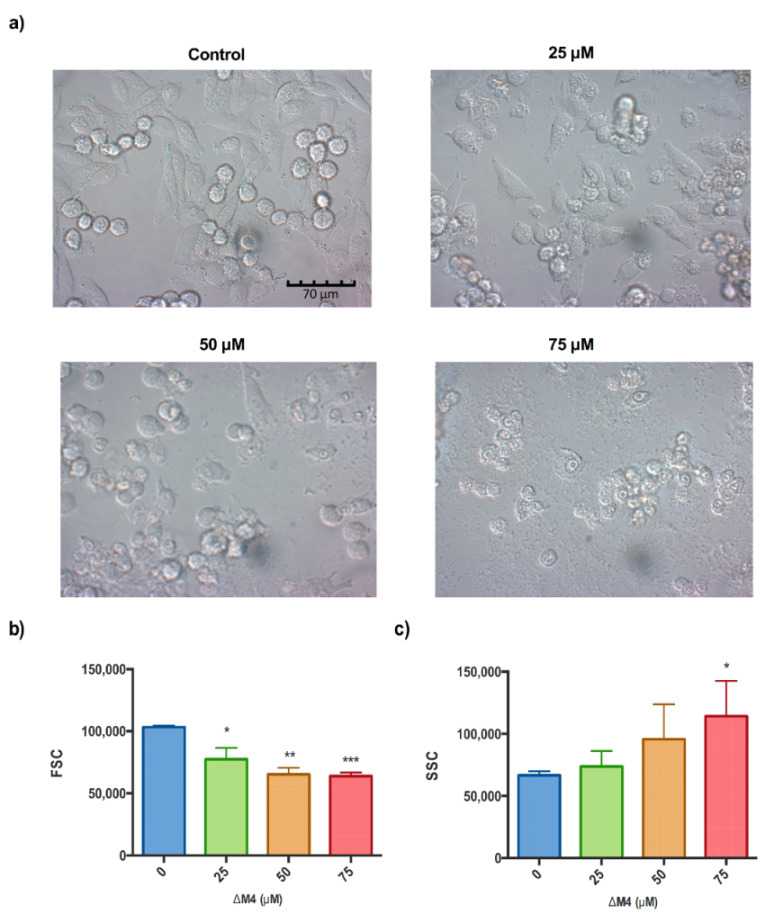
Morphological characterization of A375 cells after ΔM4 treatment. Cells were treated with different concentrations of ΔM4 for 24 h. (**a**) Direct observation by differential interference contrast (DIC) microscopy; (**b**) Measurement of the cell size represented as the mean of the intensity of the signal detected by the forward scatter (FSC) parameter by flow cytometry; (**c**) Measurement of cell granularity represented as the mean of the intensity of the signal detected by the side scatter (SSC) parameter by flow cytometry. Data are expressed as the mean ± SEM of three independent experiments. One-way ANOVA revealed the difference with respect to non-treated cells, where * *p* ≤ 0.05, ** *p* ≤ 0.01, and *** *p* ≤ 0.001.

**Figure 4 molecules-25-05684-f004:**
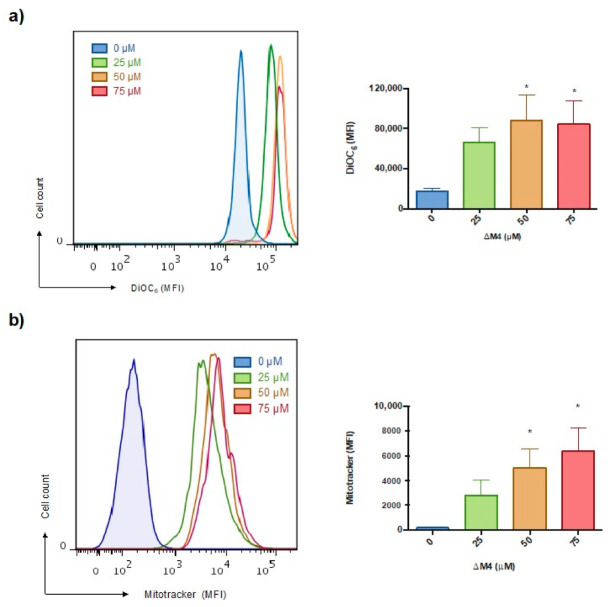
Effects of ∆M4 on the mitochondrial integrity of A375 cells. Cells were treated with different concentrations of ΔM4 for 24 h and subsequently analyzed by flow cytometry. (**a**) Mitochondrial membrane polarization was evaluated with DiOC_6_ uptake; the left panel shows a representative histogram for the increase in dye caption, and the bar graph expresses the mean ± SEM of MFI obtained in three independent experiments. (**b**) Mitochondrial ROS production quantification; the left panel shows a representative histogram for the increase in the fluorescent intensity of Mitotracker in A375 mitochondrion, and the bar graph expresses the mean ± SEM of MFI obtained in three independent experiments. One-way ANOVA revealed the difference with respect to non-treated cells, where * *p* ≤ 0.05.

**Figure 5 molecules-25-05684-f005:**
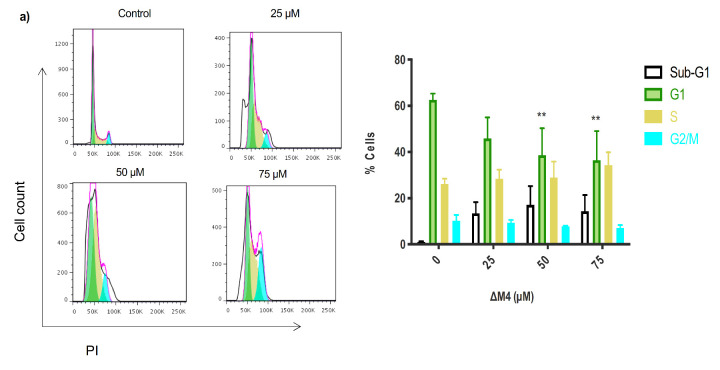
Cell cycle distribution after ∆M4 exposure in melanoma skin cancer and non-tumoral cells. Cells were treated with different concentrations of ∆M4 for 24 h. (**a**) A375, melanoma cancer cells; (**b**) HaCaT, normal human keratinocyte cell line. The following are shown for each cell line: A representative histogram of flow cytometry analysis of the cell cycle distribution; bar graphs for quantification of the cell cycle distribution of the total cell population in the different phases of the cell cycle; and two-way ANOVA for sub-G1, G1, S, and G2/M populations, displaying the difference with respect to untreated cells, where ** *p* ≤ 0.01.

**Table 1 molecules-25-05684-t001:** The half-maximal inhibitory concentration (IC_50_) values of the peptide ∆M4 on non-tumoral HaCaT cells and melanoma A375 skin cells, and the corresponding selectivity index (SX).

	HaCaT Cells	A375 Cells	Selectivity Index (SX) *
IC_50_ values (μM)	88.56	9.31	951.23

* SX value > 100 denotes that the cytotoxic effect is more selective in cancer cells.

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
