# Peer review of "Synthetic Peptide ΔM4-Induced Cell Death Associated with Cytoplasmic Membrane Disruption, Mitochondrial Dysfunction and Cell Cycle Arrest in Human Melanoma Cells"

_molecules, 2020, doi:10.3390/molecules25235684_

Round 1

Reviewer 1 Report

In the paper entitled "Synthetic Peptide ΔM4-Induced Cell Death Associated with Cytoplasmic Membrane Disruption, Mitochondrial Dysfunction and Cell Cycle Arrest in Human Melanoma Cells", the authors collected good data about the selective cytotoxicity of the synthetic peptide ΔM4, exploring its 
activity on two human skin cell lines: melanoma cells A375 and spontaneously immortalized keratinocytes (HaCaT).

The scientific approach is good and the experiments are well described and conducted, although I have some doubts that I hope the authors can easily satisfy.

  1. What is the rationale for peptide ΔM4 design? The authors should specify the characteristics of the peptide sequence and the functional relationship with the cellular target;
  2. Why do they explain the synthesis of the peptide LTX-315, if it is never experimentally used in the manuscript but only mentioned in the discussion?
  3. Has TFA been removed from the synthetic peptide? Trifluoroacetic acid is commonly present in synthetic peptide preparation and it could be toxic for cells or alter their physiology. If the authors have removed the TFA or if synthetic peptided was TFA free, they should indicate it.
  4. Is peptide ΔM4 stable? In addition to cytotoxicity assay presented, perhaps the authors should also have verified the behavior of the peptide on cells over time.   
  5. In paragraph 4.6 the authors should specify how they permeabilized the cells.
  6. It would also useful to observe an analysis of morpholigical characterization of haCaT cells after ΔM4 tratment.

Minor issues

In figure 1 (panel b) and in figure 4 (panel a and b), the axes of the histograms associated to flow cytometry have characters that are too small, practically illegible;

Many abbreviations are not explained;

Author Response

Manuscript No.: MOLECULES-988412

Title: “Synthetic Peptide ΔM4-Induced Cell Death Associated with Cytoplasmic Membrane Disruption, Mitochondrial Dysfunction and Cell Cycle Arrest in Human Melanoma Cells

  RESPONSE TO REVIEWERS

 Authors appreciate the comments and suggestions of the reviewers, which contributed to improve our manuscript. All changes made on the manuscript are detailed below:

 REVIEWER #1: The scientific approach is good and the experiments are well described and conducted, although I have some doubts that I hope the authors can easily satisfy.

  1. What is the rationale for peptide ΔM4 design? The authors should specify the characteristics of the peptide sequence and the functional relationship with the cellular target;

We appreciate the advice of the Reviewer in order to include information about ∆M4. As the paper was originally submitted to the Special Issue Hybrid Compounds, Multitarget Ligands, and Conjugate Derivatives as New Targeted Anticancer Agents, we did not included this part. The information on the design and origin of the peptide was included in the section 2.1 Design and Prediction of the 3D Structure of ΔM4 (lines 81 to 98).

  1. Why do they explain the synthesis of the peptide LTX-315, if it is never experimentally used in the manuscript but only mentioned in the discussion?

We are very sorry for the mistake. The sequence was included and it is confusing in the way it was presented. The mistake was removed from the section 4.2 Peptide Synthesis (line 347-350)

  1. Has TFA been removed from the synthetic peptide? Trifluoroacetic acid is commonly present in synthetic peptide preparation and it could be toxic for cells or alter their physiology. If the authors have removed the TFA or if synthetic peptided was TFA free, they should indicate it.

We agree with the reviewer, TFA induces uncontrolled effects on cells. For this reason we ordered the synthesis from GenScript. This company offers several options as part of the customized peptide service, including the TFA removal procedure. We include the certificate of analysis of ΔM4 received from GenScript, where is possible to corroborate the TFA removal. The modification was included in lines 347-350.

  1. Is peptide ΔM4 stable? In addition to cytotoxicity assay presented, perhaps the authors should also have verified the behavior of the peptide on cells over time.

We agree with the Reviewer, the stability of the peptide is a key factor that could affect the results. However, the peptide was bought in a lyophilized form and kept at –40 °C to prevent or minimize peptide degradation. When the peptide was prepared in solution, only the necessary amount of peptide was stored frozen in individual aliquots to avoid freeze-thaw cycles. Also, the peptide was never exposed to pHs other than 7.4. ∆M4 sequence does not contain Cys and Met residues, which are the predominant residues that undergo reversible oxidation. Finally the medium used in the biological experiments does not contain proteases that could break down the sequence. For all these reasons the stability of the peptide was not taken as part of this study, considering that all the precautions were taken (No information was included in the text).

  1. In paragraph 4.6 the authors should specify how they permeabilized the cells.

We would like to provide clarification in response to the comment of the Reviewer. Cell cultures were photographed with an inverted microscope under differential interference contrast (DIC) Cells were not therefore permeabilized, as no fixation procedure was performed to the cell cultures (No information was included in the text).

  1. It would also useful to observe an analysis of morpholigical characterization of haCaT cells after ΔM4 tratment.

As a preliminary experiment, we performed the morphological characterization of HaCaT and A375 cells after the ΔM4 treatment using flow cytometry. HaCaT cells were treated with different concentrations of ΔM4 for 24 h, after which period the cell size and cell granularity were monitored by flow cytometry as the mean of the intensity of the signal detected by the FSC parameter and as the mean of the intensity of the signal detected by the SSC parameter, respectively. Data are expressed as mean ±SEM of three independent experiments and data were analyzed using one-way ANOVA. We did not find any difference in comparison with non-treated cells. We add the result to the reviewer, for this reason the results were not included and we only present the results with ΔM4. No information was included in the text.

Minor issues

  1. In figure 1 (panel b) and in figure 4 (panel a and b), the axes of the histograms associated to flow cytometry have characters that are too small, practically illegible;

We appreciate the observation of the Reviewer, the figure was modified to improve the visualization and changed in the manuscript (lines 113-114).

  1. Many abbreviations are not explained

Thank you very much for the observation, we have carefully checked the manuscript and included all the abbreviations missing (Lines 109, 119, 274-276 and 294).

Reviewer 2 Report

This manuscript simply shows the effect of a synthetic peptide in human melanoma cells. Overall, this study needs to further investigate in order to meet the criteria to be published in Molecules. Below are some minors/major concerns for this study:

  1. What is the mechanism of the results? How did the peptide cause the increasing of ROS and the accumulation in S-phase of cell cycle?
  2. Where did the peptide locate after the treatment? Are there any evidence to show the location of the peptide?
  3. According to the discussion, hyperpolarization of the inner mitochondrial membrane can lead to the release of apoptotic factors then activate the intrinsic apoptosis pathway. Are there any evidence to show the activation of the apoptosis pathway?
  4. Did experiments carried out on the other cell line of the melanoma cells apart from A375? If you did, how does the result look like? Please clarify.

Also some recent references need to be cited as the background and/or supportive info for this manuscript. For examples:

https://doi.org/10.1016/j.semcancer.2020.08.010  

Molecular Therapy -- Oncolytics 16: 7-19

Author Response

Manuscript No.: MOLECULES-988412

Title: “Synthetic Peptide ΔM4-Induced Cell Death Associated with Cytoplasmic Membrane Disruption, Mitochondrial Dysfunction and Cell Cycle Arrest in Human Melanoma Cells

RESPONSE TO REVIEWERS

 Authors appreciate the comments and suggestions of the reviewers, which contributed to improve our manuscript. All changes made on the manuscript are detailed below:

 REVIEWER #2:  This manuscript simply shows the effect of a synthetic peptide in human melanoma cells. Overall, this study needs to further investigate in order to meet the criteria to be published in Molecules. Below are some minors/major concerns for this study:

  1. What is the mechanism of the results?

We appreciate the question of the reviewer. The experimental evidence that we obtained in this research is not enough evidence to propose a mechanism of action of ΔM4 peptide. This was the first evaluation of this sequence in human cells. We consider that further molecular biology experiments, such as determination of caspase and kinase activity, are needed to complete the full understanding of the peptide biological potential. With some recognized peptides like LL37, Magainin, Buforin and Daptomycin, several studies were necessary to understand the full view of these peptides as antimicrobial agents. Our aim is to continue the experiments in the near future, and we have already ordered the reagents to perform caspase determination, but in our country there are still restrictions on imports due to the COVID-19 pandemic regulations of the government. Therefore, for us is impossible to include these results in this publication. However, the suggestions of the Reviewer are extremely valuable for us to continue our research with ΔM4. (No information was included in the text)

How did the peptide cause the increasing of ROS and the accumulation in S-phase of cell cycle?

As we explained before, more experiments are needed to conclude and propose a mechanism of action for ΔM4. However, mitochondrial membrane potential changes were detected using DiOC6, and it is well known that dysfunctional mitochondria are a source of ROS, mainly through the electron transfer chain (ETC), which generates an escape of electrons that react with oxygen (O2) and form O2- free radicals. The dismutation of O2- results in the formation of H2O2 and HO-, which are both capable of inducing damage to critical cell structures, possibly increasing the cytotoxic effect of the peptides.

The accumulation in the S stage could be associated with DNA damage and the response to replication stress, which cause the cell to stop its normal cycle in order to attempt to repair the genetic material. However, we explained in the manuscript that further study would be necessary to test this hypothesis.

  1. Where did the peptide locate after the treatment? Are there any evidence to show the location of the peptide?

The experiments included in the publication were not focused on the determination of the peptide location. We have no evidence to respond to the Reviewer and any answer will be barely a speculation about the location of the peptide after the treatment. After this first approximation of ΔM4 in cancer cells, we have found valuable information to design the next steps in the research, including the option to label the peptide and make possible to determine the location of the peptide in the cell.

  1. According to the discussion, hyperpolarization of the inner mitochondrial membrane can lead to the release of apoptotic factors then activate the intrinsic apoptosis pathway. Are there any evidence to show the activation of the apoptosis pathway?

As we have expressed above, more experiments are needed to propose with high certainty a mechanism of action by which ΔM4 exert its antiproliferative activity. We consider that our conclusions about the potential of ΔM4 were not speculative, and we also mention in the discussion that more experiments are necessary to prove some of the hypotheses we formulate in the manuscript.

  1. Did experiments carried out on the other cell line of the melanoma cells apart from A375? If you did, how does the result look like? Please clarify.

The antiproliferative activity of ΔM4 was only evaluated in A375 cell lines and HaCaT immortalized keratinocytes. However, we have already ordered ATCC A431 (epidermoid carcinoma cells) and CRL 7762 (basal cell carcinoma cells) to continue our experiments in the exploration of the antiproliferative activity of ΔM4. We hope that the restrictions on imports will not mean more than 6-8 months will be required to obtain cancer cell lines, as was the case in the past.

  1. Also some recent references need to be cited as the background and/or supportive info for this manuscript. For examples:

https://doi.org/10.1016/j.semcancer.2020.08.010, Molecular Therapy -- Oncolytics 16: 7-19

We appreciate the observation of the Reviewer, the reference was included in the introduction, and based in the paper suggested we include two more examples of ACPs to enrich the discussion.

Reviewer 3 Report

In the present work the effect of a synthetic peptide was investigated with respect to selective activity against human melanoma cells versus a benign cell line. Although, interesting findings were made during the study, the manuscript suffers from some general issues that need to be addressed by the authors in order to justify publication in Molecules:

Line 18: usually viability of benign cells is also affected

Line 56: this is not quite true - many AMPs accumulate in the kidneys to cause nephrotoxicity - this also applies to clinical drugs like the antibiotic colistin

Line 58:  the use of the term “therapeutic peptides” might indicate clinical use - in fact no ACPs fulfil this as also stated in this review (i.e., ref. [16]): “As shown in Table 1, most cytotoxic peptides, like the antimicrobial peptides, are in pre-clinical stages of testing and have yet to enter human clinical trials for cancer”.

Line 71: what is the origin of this peptide besides being synthetic - is it an analogue of another known peptide or identified by screening of a compound library ? This question is so essential that it should be clarified in the revision

Line 91 (Figure 1.): at the dose required for effective killing of cancer cells there seems to be a significantly reduced viability of benign cells (typically a level of 80% viability should be maintained at the active concentration)

Line 99 (Table 1): a 10-fold window between the active (against cancer cells) and non-toxic (toward benign cells) concentrations does not appear to constitute a satisfactory safety margin – it would be preferred to include a comparison with a control compound that is actually used against melanoma (or other skin cancer disease – or even just a common anticancer drug)

Line 187: this entire paragraph (i.e., lines 187-199) appears to be more suitable for placement in the introduction - in a shortened version

Line 299: if termed a “control peptide” it would be expected that this compound was tested in parallel with the DM4 peptide

Line 307: LTX-315 appears not to be used in the present study - just mentioned in the discussion, but it seems

Author Response

Manuscript No.: MOLECULES-988412

Title: “Synthetic Peptide ΔM4-Induced Cell Death Associated with Cytoplasmic Membrane Disruption, Mitochondrial Dysfunction and Cell Cycle Arrest in Human Melanoma Cells

RESPONSE TO REVIEWERS

 Authors appreciate the comments and suggestions of the reviewers, which contributed to improve our manuscript. All changes made on the manuscript are detailed below:

 REVIEWER #3: In the present work the effect of a synthetic peptide was investigated with respect to selective activity against human melanoma cells versus a benign cell line. Although, interesting findings were made during the study, the manuscript suffers from some general issues that need to be addressed by the authors in order to justify publication in Molecules:

  1. Line 18: usually viability of benign cells is also affected

We appreciate the observation of the Reviewer, line 18 in the abstract was rewritten to be consistent with the purpose of the Abstract.

  1. Line 56: this is not quite true - many AMPs accumulate in the kidneys to cause nephrotoxicity - this also applies to clinical drugs like the antibiotic colistin

We agree with the Reviewer, we decided to remove this information; the principal goal was to highlight the advantages of the antimicrobial peptides (Lines 55-57).

  1. Line 58: the use of the term “therapeutic peptides” might indicate clinical use - in fact no ACPs fulfil this as also stated in this review (i.e., ref. [16]): “As shown in Table 1, most cytotoxic peptides, like the antimicrobial peptides, are in pre-clinical stages of testing and have yet to enter human clinical trials for cancer”.

We appreciate the observation of the Reviewer, the term “therapeutic peptides” is not correct. We remove this word from the Introduction. The change can be observed in the manuscript (Line 58-59).

  1. Line 71: what is the origin of this peptide besides being synthetic - is it an analogue of another known peptide or identified by screening of a compound library ? This question is so essential that it should be clarified in the revision

We appreciate the advice of the Reviewer in order to include information about ∆M4. As the paper was originally submitted to the Special Issue Hybrid Compounds, Multitarget Ligands, and Conjugate Derivatives as New Targeted Anticancer Agents, we did not include this part. The information on the design and origin of the peptide was included in the section 2.1 Design and Prediction of the 3D Structure of ΔM4 (lines 81 to 98).

  1. Line 91 (Figure 1.): at the dose required for effective killing of cancer cells there seems to be a significantly reduced viability of benign cells (typically a level of 80% viability should be maintained at the active concentration)

We agree with the Reviewer, a novel peptide or new drug should be safe for benign cells to be considered a potential therapeutic agent. However, the in vitro preliminary results of a peptide or drug can be different to the results obtained in vivo. In vitro experiments have different variables that are not included in vivo, but extremely valuable in enabling understanding of how the peptide exerts its biological activity. It is also important to mention that the potential application of ∆M4 would be in a topical administration because we are evaluating its potential in skin cancer. The fact that it is not administrated systemically in the body will reduce significantly the side effects of the peptide.

  1. Line 99 (Table 1): a 10-fold window between the active (against cancer cells) and non-toxic (toward benign cells) concentrations does not appear to constitute a satisfactory safety margin – it would be preferred to include a comparison with a control compound that is actually used against melanoma (or other skin cancer disease – or even just a common anticancer drug)

We appreciate the observation of the Reviewer. In our experimental design, it was very hard to decide the control for the experiments. The first reason was the mechanism of action; chemotherapeutics currently in use differ completely from ACPs. Additionally, the first treatment of skin cancer is not the administration of chemotherapeutics, but surgery. For this reason, we were interested in comparing our results with other results of peptides tested against melanoma cells and, for this reason LTX-315 is cited in the discussion. As can be corroborated in the website http://www.lytixbiopharma.com/news/432/130/Lytix-Biopharma-enters-milestone-agreement-with-Verrica-Pharmaceuticals-for-its-lead-candidate-LTX-315.html, Lytix BioPharma has published the following “Our lead drug candidate, LTX-315, has shown very promising efficacy and safety signals in cancer patients during Phase I/II studies and we are excited that this partnership with Verrica will expand the applications for LTX-315” .

  1. Line 187: this entire paragraph (i.e., lines 187-199) appears to be more suitable for placement in the introduction - in a shortened version

We agree with the observation of the Reviewer, the lines between 187 and 199 were reorganized and the information was reduced to look as the preamble of the Discussion.

  1. Line 299: if termed a “control peptide” it would be expected that this compound was tested in parallel with the DM4 peptide

Thank you very much for the observation of the Reviewer, the word “control”, was used incorrectly. The word was removed to avoid the misinterpretations in the Discussion (line 330).

  1. Line 307: LTX-315 appears not to be used in the present study - just mentioned in the discussion, but it seems

We are very sorry for the mistake. The sequence was included in the Material and Methods method and it is confusing in the way it was presented. The mistake was removed from the section 4.2 Peptide Synthesis (line 347-350).

Round 2

Reviewer 2 Report

I believe this reference will also able to help to support this manuscript (as a background and discussion info):

Molecular Therapy -- Oncolytics 16: 7-19

Author Response

Manuscript No.: MOLECULES-988412

Title: “Synthetic Peptide ΔM4-Induced Cell Death Associated with Cytoplasmic Membrane Disruption, Mitochondrial Dysfunction and Cell Cycle Arrest in Human Melanoma Cells

RESPONSE TO REVIEWERS

The authors appreciate the comments and suggestions of the reviewers, which contributed to improve our manuscript. All changes made to the manuscript are detailed below:

 REVIEWER #2:  I believe this reference will also able to help to support this manuscript (as a background and discussion info):

Molecular Therapy -- Oncolytics 16: 7-19

We agree with the Reviewer, the reference helps to support the manuscript and it was included in the Introduction (lines 51-52 and 58-62) and the Discussion (250 - 260).

We apologize for the mistake, we assumed that the https://doi.org/10.1016/j.semcancer.2020.08.010 was the same reference and we did not check the information. We are very sorry, we appreciate the inclusion of this reference in the manuscript. The changes can be followed in red colour.

Reviewer 3 Report

The Authors have responded satisfactorily to the comments made during review.

Author Response

The author appreciate your observations and suggestions to improve the manuscript